# FDG-PET/CT and Para-Aortic Staging in Endometrial Cancer. A French Multicentric Study

**DOI:** 10.3390/jcm10081746

**Published:** 2021-04-17

**Authors:** Camille Sallée, François Margueritte, Sébastien Gouy, Antoine Tardieu, Jérémie Belghiti, Eric Lambaudie, Pierre Collinet, Frédéric Guyon, Maxime Legros, Jacques Monteil, Tristan Gauthier

**Affiliations:** 1Department of Gynaecology and Obstetrics, CHU Limoges, 87042 Limoges, France; fmargueritte@gmail.com (F.M.); antoine.tardieu@chu-limoges.fr (A.T.); legros.maxime12@gamil.com (M.L.); tristan.gauthier@chu-limoges.fr (T.G.); 2Department of Surgery, Gustave Roussy Comprehensive Cancer Center, 94800 Villejuif, France; sebastien.gouy@gustaveroussy.fr; 3Department of Gynecologic and Breast Surgery and Oncology, Pitié-Salpêtrière, AP-HP, 75013 Paris, France; jeremie.bleghiti@aphp.fr; 4Institut Paoli Calmettes and CRCM, 13009 Marseille, France; lambaudiee@ipc.unicancer.fr; 5Gynaecological Surgery Unit, Jeanne de Flandre Hospital, University Hospital of Lille, 59000 Lille, France; pierre.collinet@chru-lille.fr; 6Institut Bergonié, 229, 33000 Bordeaux, France; f.guyon@bordeaux.unicancer.fr; 7Nuclear Medicine Department, Limoges University Hospital, 87042 Limoges, France; jacques.monteil@chu-limoges.fr; 8UMR-1248, Faculté de Médecine, 87000 Limoges, France

**Keywords:** endometrial cancer, FDG-PET/CT, lymphadenectomy, para-aortic, high risk

## Abstract

Background: FDG-PET/CT is a noninvasive examination that could be helpful for the management of endometrial cancer. The aim of this study was to evaluate the performance of FDG-PET/CT in assessing para-aortic lymph-node involvement in high-risk endometrial cancer. Methods: We performed a retrospective multicenter study including all patients who had a high-risk endometrial cancer with a preoperative FDG-PET/CT and a para-aortic lymphadenectomy (PAL) between 2009 and 2019. The main objective was to evaluate the overall performance of FDG-PET/CT. The secondary objectives were to evaluate its performances according to the histological type and according to FDG-PET/CT date (before or after hysterectomy), and to compare its overall performance with that of the MRI scan. Results: We included 200 patients from six different centers. After the false positive FDG-PET/CT was reread by nuclear physicians, FDG-PET/CT had a sensitivity of 61.8%, a specificity of 89.7%, a positive predictive value of 69.4%, a negative predictive value of 86.1%, and an AUC of 0.76. There were no statistically significant differences in the performances according to either histological type and or FDG-PET/CT date. The sensitivity of FDG-PET/CT was better than that of MRI (*p* < 0.01), but the specificity was not (*p* = 0.82). Conclusion: Currently, FDG-PET/CT alone cannot replace PAL for the lymph node evaluation of high-risk endometrial cancers. It seems essential to reread it in multidisciplinary meetings before validating the therapeutic management of patients, particularly in the case of isolated para-aortic involvement.

## 1. Introduction

Endometrial cancer is the fourth most common cancer in women with more than 8000 cases per year in France [1]. It is currently necessary to know the lymph node status and the FIGO stage in order to establish the most appropriate therapeutic strategy. Indeed, the PORTEC-3 study found a gain in recurrence-free survival for stage III endometrial cancers when chemotherapy is added to radiotherapy (69.3% vs. 58.0%, *p* = 0.032) justifying knowledge of lymph node status [2]. The risk of lymph node involvement is strongly correlated with the characteristics of the tumor and depends on its histological type, grade, myometrial invasion, and the presence of lymphatic emboli [3].

Para-aortic lymphadenectomy (PAL) is a surgical procedure with a non-negligible morbidity rate, particularly in an elderly population with comorbidities [4]. Its therapeutic impact on survival is currently a source of debate [5]. Knowledge of the pelvic and para-aortic lymph node status influences the therapeutic decision through the indication or not of more or less extensive radiotherapy and systemic treatment [6]. Lymphadenectomy is currently the “gold standard” for the evaluation of lymph node involvement in endometrial cancer.

The 18F-fluorodeoxyglucose positron emission tomography–computed tomography (FDG-PET/CT) is a noninvasive examination that could change the management of endometrial cancer and is listed in the European Society of Gynecological Oncology (ESGO) recommendations as an option [7]. It is already widely used in the management of locally advanced cervical cancer [8]. Similarly, in cases of endometrial cancer with indications for para-aortic lymphadenectomy, performing an FDG-PET/CT scan could be helpful in deciding whether or not to perform a lymphadenectomy. Currently, only a few studies have evaluated its interest in the lymph node staging of endometrial cancers.

A previous single-center study found 100% specificity and a positive predictive value with a 92.9% accuracy rate for FDG-PET/CT for the evaluation of para-aortic involvement, particularly in high-risk histological type 1 endometrial cancers [9]. FDG-PET/CT performance was lower in type 2, with an increased risk of false negatives [9]. However, the results were not significant due to a low number of patients and a lack of power and need to be re-evaluated in a larger population.

In order to check the results of our initial series, we carried out a new, multicenter study, allowing us to recruit a larger number of patients.

The main objective of our study was to investigate the performance of FDG-PET/CT in the evaluation of para-aortic involvement in high-risk endometrial cancers. The secondary objectives were to evaluate its performance according to the histological type but also according to FDG-PET/CT date (at diagnosis or after hysterectomy) and to compare it with that of MRI.

## 2. Materials and Methods

The study was conducted according to the guidelines of the Declaration of Helsinki and approved by the Ethics Committee of CHU Limoges (n° 311–2019–77, 15 May 2019).

### 2.1. Population

We conducted a retrospective multicenter study evaluating the performance of preoperative FDG-PET/CT in assessing para-aortic involvement in high-risk and advanced endometrial cancers between 1 January 2009 and 31 December 2019.

The six centers participating in the study were made up of three University Hospitals (Lille, Limoges and Paris) and three cancer centers (Bordeaux, Marseille and Villejuif).

### 2.2. Inclusion Criteria

We included over-18-year-old patients with high-risk type 1 or type 2 endometrial cancer who had undergone preoperative FDG-PET/CT or FDG-PET/MRI (in one center) and surgery including para-aortic lymphadenectomy.

Para-aortic lymphadenectomy was performed from the left renal vein (upper limit) to the iliac bifurcations including the promontory and common iliacs (lower limit). Nodes at the internal, external and obturator iliac levels were considered to part of the pelvic lymphadenectomy.

### 2.3. Exclusion Criteria

The exclusion criteria were absence of preoperative FDG-PET/CT, PAL in case of recurrence, PAL in case of neo-adjuvant chemotherapy and histological uterine sarcomas.

### 2.4. Data Collection

Patient data were collected from the paper and/or computerized patient records at each center. 

The following data were collected:Patient characteristics: date of birth, date of diagnosis, BMI and ASA score prior to lymph node surgery.Baseline data: initial histology, initial grade, MRI results of para-aortic involvement (size of pelvic and para-aortic nodes), preoperative FDG-PET/CT results (maximum SUV of tumor and pelvic and para-aortic nodes and their size) and stage.Initial indication for PAL: at the same time as hysterectomy, after hysterectomy results or after a positive pelvic lymph node finding (lymphadenectomy or sentinel node (SN)).Node surgery: date and approach (laparoscopy, laparotomy and/or robotic surgery).Post-operative anatomopathological results: peritoneal cytology, definitive histology, grade, presence of emboli, presence of hormonal receptors, tumor size, presence of pelvic and/or para-aortic lymph node metastases and their maximum size, final FIGO stage.

FDG-PET/CT was considered positive in the para-aortic area according to the nuclear physician’s interpretation and report, which was approved by the Tumor Board Meeting (TBM).

All FDG-PET/CT scans that were false positive in the para-aortic area were reviewed a posteriori by the nuclear physicians of each center.

For the anatomopathological results, para-aortic nodes were considered positive whether there were micro- or macrometastases. Pathology analysis was performed according to the standard procedure, with hematoxylin-eosin-saffron staining.

### 2.5. Aims of the Study

The main objective was to assess the diagnostic performance of FDG-PET/CT in the evaluation of para-aortic involvement in high-risk endometrial cancers.

The secondary objectives were to assess these performances according to the histological type of the tumor, but also according to the surgical sequence of the lymphadenectomy (initial or restaging).

Another objective was to compare the diagnostic performance of FDG-PET/CT with that of MRI.

### 2.6. Statistical Analysis

Quantitative data were expressed as means, standard deviations and extremes, and qualitative data as percentages. The diagnostic performance of FDG-PET/CT and MRI was evaluated according to sensitivity, specificity, positive predictive value, negative predictive value and AUC. The “gold standard” of this evaluation was the definitive anatomopathological examination of the para-aortic lymph nodes.

For continuous variables the comparison was made using Student’s t-test and for categorical variables Fisher’s exact test or a Chi2 test. A McNemar, Chi2 or Fisher’s exact test to compare the intrinsic characteristics of a test (depending on whether or not the samples were independent) was used.

All the analyses and calculations were done with the STATA 15.1 IC^®^ software (StataCorp LLC, College Station, TX, USA). A *p*-value < 0.05 was considered statistically significant.

## 3. Results

### 3.1. Patient Characteristics

From January 2009 to December 2019, 200 patients were treated for high-risk endometrial cancer and benefited from PAL with preoperative FDG-PET/CT in six French centers (Figure 1). Out of the 200 patients, 13 received FDG-PET/MRI instead of FDG-PET/CT.

Only one person refused to allow her data to be used in our study.

The characteristics of the patients are presented in Table 1. The mean age and BMI were 62.72 years (27–83) and 27.90 kg/m^2^ (15.43–50.17), respectively. With regard to preoperative histological types, 96 patients had histological type 1 (48%). This was grade 3 in 54.00% of cases. In total, 57.00% of patients had advanced disease suspected on MRI and/or FDG-PET/CT (stage ≥ III). The mean interval was 1.2 months (0–5) between diagnosis and hysterectomy and 2.3 months (1–5) between diagnosis and secondary PAL. PAL was performed at the same time as hysterectomy in 74.50% of cases. Surgery was preferably performed through a minimally invasive route (56%).

Definitive histological data as well as post-operative FIGO stages are summarized in Table 2. Pelvic lymphadenectomy was not performed for 19 patients (9.5%). With regard to histological type, 108 patients (54.00%) had type 1 and 92 patients (46%) had type 2. Para-aortic lymphadenectomy was positive in 55 patients (27.5%). Peritoneal cytology was performed in 55% of cases.

### 3.2. Main Objective

The initial false positive (FP) rate was 10% (20/200) and the false negative rate was 10.5% (21/200). There was no significant difference in the number of FPs between centers (*p* = 0.19). There were 65% type 1s in this subgroup and 80% of PALs were performed immediately, at the same time as the hysterectomy. Two patients received FDG-PET/MRI instead of FDG-PET/CT. 

After rereading by the nuclear physicians at each center, the FDG-PET/CT finally did not present an abnormal uptake in the lymph node area for two patients, but it was probably a urinary artifact. For three patients, the increased FDG-PET/CT uptake was in the iliac area (primitive, internal or external) and was therefore considered as pelvic nodes by the nuclear physician who reviewed the FDG-PET/CTs. For one patient, only three lymph nodes were removed during PAL. For another patient, the only para-aortic fixation node was behind the left renal vein. For two patients, we were unable to obtain a review because the FDG-PET/CTs had been performed externally, without access to the original examination data. If we remove the three patients for whom abnormal uptake was in the iliac area and the two patients who finally had no lymph node involvement on FDG-PET/CT, the final FP rate is 7.5% (15/200).

After rereading, the FDG-PET/CT performance for the evaluation of para-aortic involvement showed a sensitivity (Se) of 61.8%, a specificity (Sp) of 89.7%, a PPV of 69.4%, an NPV of 86.1% for an AUC of 0.76, whatever the histological type. 

After pathological analysis, 55 patients had para-aortic involvement. Of these, five had no associated pelvic lymphadenectomy and 14 (25.45%) had isolated para-aortic involvement.

### 3.3. FDG-PET/CT Performance According to Histological Type

In our study, 108 patients (54%) had histological type 1 on the final histology. PAL was positive in 21.3% of the cases. Se, Sp, PPV and NPV, as well as FDG-PET/CT AUC in this subgroup were 65.2%, 88.2%, 60%, 90.4%, and 0.77, respectively (Figure 2).

For histological type 2, 92 patients (46.00%) were involved. PAL was positive in 34.78% of the cases. The FDG-PET/CT performance was 59.4%, 91.7%, 79.2%, 80.9%, and 0.76, respectively (Figure 2).

No statistically significant difference was found between these two groups in terms of Se (*p* = 0.66) and Sp (*p* = 0.50) of the FDG-PET/CT.

### 3.4. FDG-PET/CT Performance According to Its Date

For 149 people (74.5%), PAL was performed at the same time as the hysterectomy. For the remaining 51 people (25.5%), PAL was performed a second time after the hysterectomy for restaging. FDG-PET/CT was performed before the hysterectomy for 162 patients (81%). In the remaining 38 cases (19%), FDG-PET/CT was performed between the hysterectomy ± pelvic lymphadenectomy or pelvic SN and PAL.

The performance of FDG-PET/CT, when performed prior to hysterectomy, was 67.4% Se, 88.3% Sp, 72.1% PPV, 85.8% NPV, and 0.78 AUC.

The performance of FDG-PET/CT, performed after the hysterectomy, was 33.3%, 92.9%, 50.0%, 86.7%, and 0.63, respectively.

There was no difference between the two groups in terms of AUC (*p* = 0.12), specificity (*p* = 0.32) or sensitivity (*p* = 0.06).

### 3.5. Comparison of FDG-PET/CT and MRI Performances

We also compared the performance of FDG-PET/CT and MRI in the evaluation of para-aortic involvement in high-risk endometrial cancers (Table 3). The difference was statistically significant for sensitivity (*p* < 0.01) and AUC (*p* < 0.01). There was no significant difference in specificity between the two examinations (*p* = 0.82).

## 4. Discussion

### 4.1. Overall Performance

According to our study and after rereading, the performance of the FDG-PET/CT for the para-aortic evaluation of high-risk endometrial cancers is 61.8% for sensitivity, 89.7% for specificity, with a positive predictive value of 69.4%, a negative predictive value of 86.1%, and an AUC of 0.76. To the best of our knowledge, our study shows the highest number of PALs, including the highest number of positive PALs.

We were surprised by the initial FP rate of 10%, which was comparable across centers. Indeed, according to the new recommendations of the French Society of Gynecological Oncology (SFOG), 10% of patients would have been wrongly irradiated in the para-aortic area if no PAL had been performed [6]. We therefore decided to have these FDG-PET/CT reread by the nuclear physician at each center. In one center, there were two FDG-PET/MRI to be reread.

After discussion with the different nuclear physicians and by looking at the data in the literature, we decided to include in our study the 13 patients who had had FDG-PET/MRI instead of FDG-PET/CT. Indeed, FDG-PET/MRI does not appear to be inferior to FDG-PET/CT. It would even appear to be superior in the evaluation of lymph node involvement according to the studies, with better sensitivity and specificity [10,11]. However, these studies are few and lack power with fewer than 50 patients.

After rereading, there are several reasons for this FP rate. On the one hand, for one patient, only three para-aortic lymph nodes were removed during surgery. The abnormal uptake in the lymph node on FDG-PET/CT may not have been removed during PAL. On the other hand, for three patients, the FDG-PET/CT only presented an increased uptake at the level of the iliac arteries, whether primitive or external, with no increased uptake in front of the aorta or vena cava. In the French centers, primitive iliac lymphadenectomy belongs to the PAL group and internal and external iliac lymphadenectomy belongs to the pelvic lymphadenectomy group (PL). These are close anatomical areas and the FDG-PET/CT increased uptake in lymph nodes in the primary iliac area may have been removed in the pelvic lymphadenectomy at the time of surgery. In addition, for one patient, the only FDG-PET/CT increased uptake in a lymph node was located behind the left renal vein. This is the upper limit of the PAL. It is therefore possible that the node with an abnormal uptake was not removed at the time of the PAL. For five patients, the positivity was doubtful and nevertheless suspicious in the context of cancer. Finally, for two patients, the rereading did not find any abnormal uptakes in the lymph nodes. If we remove the three patients for whom the FDG-PET/CT abnormal uptake was in the iliac area and the two patients who finally had no lymph node involvement on FDG-PET/CT, the final FP rate is 7.5% (15/200). For two patients, we were not able to have a review by a nuclear physician.

The results found in our study are comparable to those found in the literature, presented in Table 4 synthesizing the different series that specifically focused on the performance of the FDG-PET/CT scan in para-aortic evaluation.

The other studies are less powerful than ours because the number of subjects and the percentage of lymph node involvement are relatively low. Positive and negative predictive values cannot be compared because they depend on the prevalence of the disease. However, the numbers of positive PALs are low in all studies.

Our data are close to those of Atri et al. which is the study with the highest number of PAL positive subjects after ours [20]. However, the FP rate in this study is not known.

In order to compensate for the number of FPs of FDG-PET/CT, Gouy et al. in their study of cervical cancers, recommend continuing to perform PAL when the FDG-PET/CT is only positive in the para-aortic area without any increased uptake in the pelvic area because the rate of isolated para-aortic lymph node metastases is very low [8]. This strategy could also be used in high-risk endometrial cancers where the rate of isolated para-aortic lymph node metastases is also low (0–7%) [22,23]. However, in our study, four patients among the 15 FPs (26.7%) had isolated para-aortic fixation on FDG-PET/CT.

It should be noted that in our study we had 14 patients with proven isolated para-aortic involvement, i.e., 25.45% of the patients with a positive PAL. These were 11 patients with type 2 and three patients with type 1. One of the reasons for this high rate is that we had a large number of histological type 2 patients (46%). Yet in most studies the percentage of type 2 is less than 20% [23,24,25]. However, to our knowledge, there are no studies in the literature comparing the rate of isolated para-aortic involvement according to histological type.

Only one retrospective study from 2014 finds results similar to ours, with 16.7% isolated para-aortic involvement [26]. In this study, most patients had lymphatic emboli (57.1%) and more than 50% myometrial involvement (81%), but there was a majority of type 1 (83.3%) [26].

As far as the FN rate is concerned, it was 10.50% in our study, confirming the fact that PAL must be carried out in the case of a negative FDG-PET/CT in order to avoid undertreating patients with para-aortic involvement. Indeed, the performance of the FDG-PET/CT in lymph node evaluation depends on the size of the metastatic nodes. Park et al. found a low diagnostic value of FDG-PET/CT for metastatic nodes less than 1 cm in size in the minor axis [21]. Similarly, Kitajima et al. evaluate the sensitivity of the FDG-PET/CT according to the size of the metastatic nodes. This increases from 16.7% for 2–4 mm nodes, to 93.3% for 10–12 mm nodes [14]. Unfortunately, in our study, the information on the size of para-aortic metastatic lymph nodes in histopathology was too heterogeneous and did not allow us to evaluate the performance of FDG-PET/CT according to lymph node size.

Since 2010 and the SENTI-ENDO study published in 2011 [27], sentinel node biopsy (SN) in low and intermediate risk endometrial cancers has become widespread. Since then, two new trials have been published evaluating the sentinel node procedure in high-risk endometrial cancers (FIRES and SHREC), in which the authors concluded that lymphadenectomy could be replaced by the sentinel node technique, insisting that this procedure should be performed by experienced surgeons [28,29]. In both studies, the rate of sentinel node complications was almost zero [28,29]. Ongoing studies on SN in high-risk situations (SENTIRAD) will enable us to provide a more precise answer to the frequency of isolated para-aortic involvement.

On the basis of these articles and our study, a combination of the sentinel node technique and FDG-PET/CT could be an alternative to PAL in patients with high-risk endometrial cancer, particularly in those with significant comorbidities and in whom PAL is particularly risky.

Moreover, in a preliminary study carried out at the Limoges University Hospital, the FDG-PET/CT has enabled the discovery of distant metastatic disease in 26.6% of patients with high-risk endometrial cancer [9]. This examination has the advantage of looking at the whole body, unlike MRI which focuses on a given region. It thus allows a distant assessment of the extent of the disease. The meta-analysis of Kakhi et al., based on seven studies, found a FDG-PET/CT sensitivity of 95.7% and a specificity of 95.4% in the search for distant metastases [30].

### 4.2. FDG-PET/CT Performance According to Histological Type

To our knowledge, only the study carried out by Legros et al. at the Limoges University Hospital concerned the performance of FDG-PET/CT according to histological type. In this study, they found a net decrease in the sensitivity and the AUC of the FDG-PET/CT for type 2 which fell from 66.7% to 45.5% and from 0.83 to 0.73, respectively, without these results being significant (*p* = 0.50 and 0.56, respectively) but the number of subjects in the study was low [9].

We did not confirm these results in our study. Sensitivity and specificity were similar between the two groups, i.e., respectively, 65.2% vs. 59.4% (*p* = 0.66) and 88.2% vs. 91.7% (*p* = 0.50) between types 1 and 2. However, our results seem to be more interpretable due to the greater number of patients and the absence of imbalance between the two groups, with 108 type 1 and 92 type 2 endometrial cancers.

### 4.3. FDG-PET/CT Performance According to Its Date

Likewise, to our knowledge, no study has been done on the performance of FDG-PET/CT according to its time of performance, before or after the hysterectomy. Indeed, in the other studies, FDG-PET/CT was always performed before the hysterectomy.

In our study, FDG-PET/CT was carried out from the initial extension assessment in 81% of cases. In the other cases, it was carried out between the initial surgery (hysterectomy ± PL or pelvic SN) and the restaging PAL. The sensitivity seems to be reduced when the FDG-PET/CT scan is performed after the hysterectomy from 67.4% before to 33.3% after the hysterectomy without these results being significant (*p* = 0.06). This difference in sensitivity can be partly explained by the large difference in the number of patients between the two groups (162 vs. 38).

### 4.4. Comparison with MRI

In accordance with the literature, we found a better diagnostic performance of FDG-PET/CT compared to MRI in the evaluation of para-aortic involvement [31,32]. The difference was statistically significant in terms of sensitivity and AUC.

Recently, Fasmer et al. compared FDG-PET/CT and diffusion MRI initial tumor markers in the prediction of lymph node involvement in endometrial cancers with significant superiority of FDG-PET/CT markers [33]. In this study, a metabolic tumor volume >27 mL on FDG-PET/CT increased the sensitivity of the examination in the prediction of lymph node involvement (81% vs. 50%) although the difference was not significant (*p* = 0.13) [33]. However, the measurement of metabolic tumor volume is not systematic and is not standardized, which can lead to interobserver and intercenter variability.

### 4.5. Strengths of the Study

The strong points of our study are the large number of patients included, the largest in the literature on the subject to our knowledge, but also its multicenter design. With 200 patients from six different French centers having had preoperative FDG-PET/CT and PAL, we can consider our results as interpretable and representative of the general population. 

We also have a good distribution of the different histological types with 54% type 1 and 46% type 2. This is not the case in the other two large studies found in the literature, where the percentage is not known in the study by Atri et al. and is 85% type 1 in the study by Park et al. [20,21]. This brings more power to our study for the comparison of FDG-PET/CT performance according to histological type.

Another strong point of our study is the number of para-aortic involvement with 55 positive PALs in the 200 patients with high-risk endometrial cancer studied (27.5%).

### 4.6. Limitations of the Study

One of the weaknesses of our study is its retrospective nature and its long period of time. We therefore have a number of missing data, notably the size of lymph node metastases, which seems to be an important criterion in the assessment of the FDG-PET/CT [14]. Additionally, the criteria to define a positive node on FDG-PET/CT were subject to the interpretation of the nuclear physicians. Moreover, improvements in PET models may have changed the detection power of lymph nodes.

Another weakness is that the reviewers of the FPs were aware of the results of PAL.

## 5. Conclusions

Currently, FDG-PET/CT alone cannot replace PAL for the lymph node evaluation of high-risk endometrial cancers.

The systematic practice of PAL is likely to be modified with the generalization of the sentinel node biopsy. However, the impact of this practice on the long-term survival of patients with high-risk endometrial cancer is not yet known.

As far as FDG-PET/CT is concerned, it seems to be essential to reread it in multidisciplinary meetings before validating the therapeutic management of patients, particularly in the case of isolated para-aortic involvement.

## Figures and Tables

**Figure 1 jcm-10-01746-f001:**
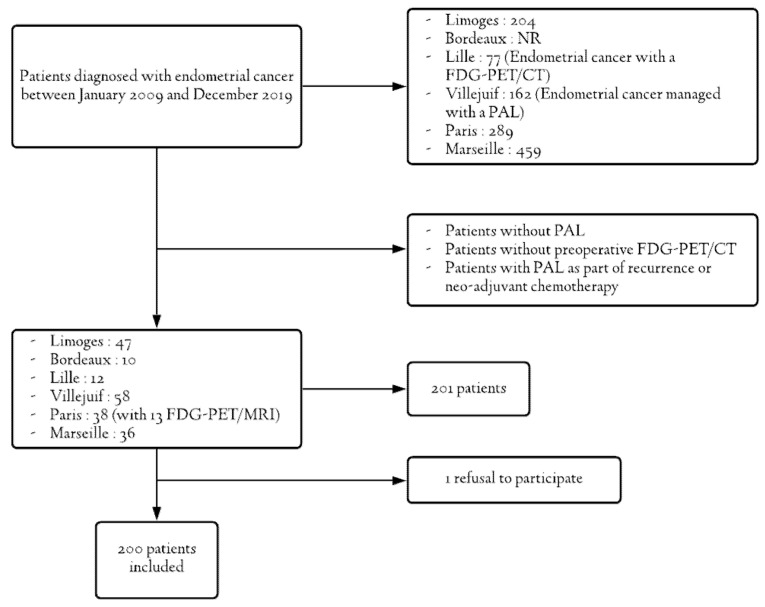
Flowchart.

**Figure 2 jcm-10-01746-f002:**
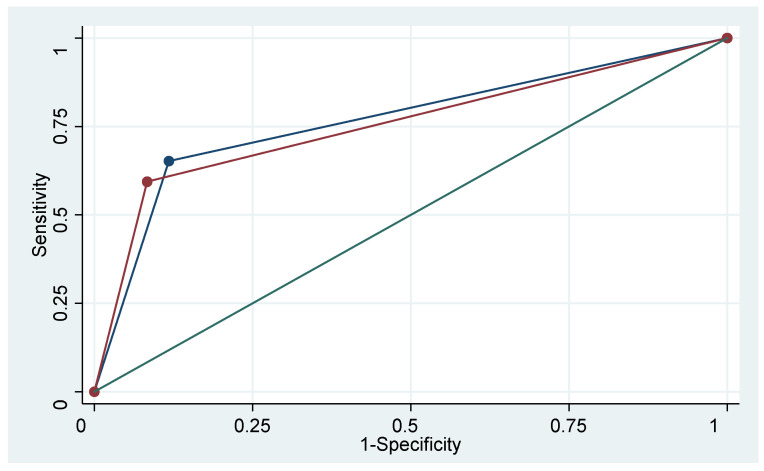
FDG-PET/CT performance according to the histological type.

**Table 1 jcm-10-01746-t001:** Patient characteristics.

Patient Characteristics	Mean ± Standard Deviation (Extremes) or *n* (%) (*N* = 200)
Age at diagnosis (years)	62.72 ± 9.19 (27–83)
BMI (kg/m^2^)	27.90 ± 6.31 (15.43–50.17)
Score ASA ^3^	
1	15 (7.50)
2	15 (7.50)
3	14 (7.00)
NR ^1^	72 (36.00)
Initial histological type	
Endometrioid	96 (48.00)
Clear cells	16 (8.00)
Papillary serous	44 (22.00)
Carcinosarcoma	26 (13.00)
Poorly differentiated	16 (8.00)
In situ	1 (0.50)
Hyperplasia with atypia	1 (0.50)
Initial grade	
1	25 (12.50)
2	48 (24.00)
3	108 (54.00)
NR	19 (9.50)
Initial stage ^2^	
IA	37 (18.50)
IB	34 (17.00)
II	15 (7.50)
IIIA	10 (5.00)
IIIB	3 (1.50)
IIIC1	37 (18.50)
IIIC2	59 (29.50)
IVA	2 (1.00)
IVB	3 (1.50)
Initial indication for PAL ^4^	
At diagnosis	149 (74.50)
After hysterectomy results	42 (21.00)
After positive pelvic lymph node finding	9 (4.50)

^1^ NR: not reported, ^2^ Stage evaluated by MRI and/or FDG-PET/CT, ^3^ ASA: American Society of Anesthesiologists, ^4^ PAL: Para-Aortic Lymphadenectomy.

**Table 2 jcm-10-01746-t002:** Final histological data.

Post-Operative Data	Mean ± Standard Deviation (Extremes) or *n* (%) (*N* = 200)
Peritoneal cytology	
Positive	18 (9.00)
Negative	92 (46.00)
NP ^1^	90 (45.00)
Final histology	
Endometrioid	108 (54.00)
Clear cells	19 (9.50)
Papillary serous	41 (20.50)
Carcinosarcoma	32 (16.00)
Grade	
1	13 (6.50)
2	52 (26.00)
3	132 (66.00)
NR ^2^	3 (1.50)
Emboli	
Yes	100 (50.00)
No	85 (42.50)
NR	15 (7.50)
Hormonal receptors	
Positive	82 (41.00)
Negative	63 (31.50)
NR	55 (27.50)
Tumor size (mm)	51.46 ± 30.59 (1; 180)
Number of patients with para-aortic lymph node involvement	55 (27.50)
Number of PA lymph nodes removed at PAL ^3^	20.33 ± 13.56 (1; 109)
Number of positive lymph nodes	4.73 ± 6.14 (1; 26)
Number of patients with pelvic lymph node involvement	60/181 (33.15)
Number of pelvic lymph nodes removed	15.34 ± 7.54 (2; 52)
Number of positives lymph nodes	2.90 ± 2.91 (1; 21)
Final FIGO ^4^ stage:	
IA	43 (21.50)
IB	23 (11.50)
II	23 (11.50)
IIIA	17 (8.50)
IIIB	7 (3.50)
IIIC1	22 (11.00)
IIIC2	46 (23.00)
IVA	5 (2.50)
IVB	14 (7.00)

^1^ NP: not performed, ^2^ NR: not reported, ^3^ Para-Aortic Lymphadenectomy, ^4^ FIGO: International Federation of Gynecology and Obstetrics.

**Table 3 jcm-10-01746-t003:** Comparison of FDG-PET/CT and MRI performances in the para-aortic area.

Performances	Sensitivity (%)	Specificity (%)	PPV (%)	NPV (%)	**AUC**
FDG-PET/CT	61.8	89.7	69.4	86.1	0.76
MRI	26.5	89.5	48.1	76.8	0.58
*p*-value	<0.01	0.82			<0.01

**Table 4 jcm-10-01746-t004:** Main studies evaluating the performance of FDG-PET/CT at the para-aortic level.

Studies	Number of Patients	Number of PAL	Number of Positive PAL	Se (%)	Sp (%)	PPV (%)	NPV (%)
Chao et al., 2006 [12]	49	NR	13	85	95	84.6	95.2
Suzuki et al., 2007 [13]	30	19	1	0	100	0	94.7
Kitajima et al., 2008 [14]	40	34	NR	51.7	99.4	83.3	97.3
Park et al., 2008 [15]	53	31	7	57.1	87.5	57.1	87.5
Suga et al., 2011 [16]	30	15	4	100	100	100	100
Crivellaro et al., 2013 [17]	76	15	6	85.7	96	87.5	96.3
Gholkar et al., 2014 [18]	20	13	1	100	66.7	20	100
Mayoral et al., 2016 [19]	13	12	3	33	88.9	50	80
Atri et al., 2017 [20]	215	160	23	65	88.0	48.4	93.8
Park et al., 2017 [21]	362	118	11	18.2	98.1	92.1	50
Legros et al., 2019 [9]	81	35	14	50	100	100	75
Our study	200	200	55	61.8	89.7	69.4	86.1

## Data Availability

Not applicable.

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
