# Peer review of "FDG-PET/CT and Para-Aortic Staging in Endometrial Cancer. A French Multicentric Study"

_jcm, 2021, doi:10.3390/jcm10081746_

Round 1
Reviewer 1 Report
JCM – 1134151
In their manuscript entitled “FDG-PET/CT and para-aortic staging in endometrial cancer. A French multicenter study,” Sallee and colleagues report a retrospective, multicenter analysis of FDG-PET/CT performance for detection of para-aortic lymph node metastases in patients undergoing hysterectomy for high risk endometrial carcinoma. They compare sensitivity and specificity to that of the gold standard pathology on lymph node dissection, and also compare the performance of FDG-PET/CT to that of MRI. The current study is an expansion of a previously reported single institution analysis that showed very high 100% specificity and 92% accuracy in patients with high risk type I endometrial cancer, but limited statistical power given relatively low numbers. Analysis of the current expanded cohort shows that FDG-PET/CT is moderately sensitive and highly specific for LN metastases in this patient population, and performs slightly better than MRI. This is more consistent with existing literature, including prospective cooperative group phase II study by the ACRIN network. Inclusion of a large percentage of patients with type two endometrial cancers (non-endometrioid) contributes useful information to the literature, and, as the authors point out, this is the largest series to date assessing FDG-PET/CT for PALN involvement from high risk endometrial cancer albeit retrospective. An additional strength includes a posteriori review of the false-positive scans by expert nuclear medicine physicians. Points for improvement are discussed below.
1. Were included patients sequential (was pre-operative FDG-PET routine for endometrial cancer at these centers), or were patients pre-selected for PET scan evaluation based on features associated with a higher risk of PA-LN involvement? This bias might explain the high percentage of PA-LN in this study (compared to the prospective ACRIN study by Atri et al, for instance).
2. The number of positive PA-LNs (average almost 5) seems quite high, perhaps indicating a population biased towards more advanced disease as mentioned in point one.
3. A posteriori review of false-positive PA-LN by nuc med physician of that center – can you clarify that the reviewers knew that these exams were false-positive, or were they reviewing blindly? This might impact interpretation of the final FP rate reported of 7.5% after omitting patients with pelvic LN (instead of PA) and no uptake on re-review.
4. What was the time interval between FDG-PET/CT(or MRI) and hysterectomy (mean or median and range)? This should be reported separately for the patients who had PA-LAD as a second procedure following hysterectomy. The authors state that their study is more powerful than others (including prospective trials); however, this information is important.
5. Additionally for patients who had delayed PA-LN, was there a difference in histologic subtype and stage (i.e. more commonly patients with stage I, type I tumors?)
6. Do the authors have information about the size of pathological nodal deposits? This information would be useful to understand if the sensitivity of the test was lower because of small-volume disease below the level of detection of FDG-PET/CT, or rather was not sufficiently concentrating of tracer to detect.
Minor Points:
1. Figure 1 presented in the manuscript draft is en Francais. In keeping with the rest of the manuscript it would be helpful to provide the figure English as well. Also including total numbers at the top or bottom of each box would be helpful for the reader.
Reviewer 2 Report
There have been many studies in the past examining the paratracheal lymph node metastatic potential of FDG-PET/CT. The present study was a retrospective multicenter study that included a relatively large number of patients with high-risk endometrial cancer who underwent preoperative FDG-PET/CT and para-aortic lymphadenectomy (PAL). However, a long-term study of 10 years may raise serious doubts about the results of the study due to differences in models and methods of operation. Preferably, various methods should be used to evaluate FDG accumulation in the primary tumor and in the lymph nodes. For example, TLG and texture analysis. In addition, the definition of positive lymph nodes on the image is unclear. If possible, a kappa test between independent examiners is needed. The definition of lymph node in pathology is also unclear. It is unclear whether micrometastases were considered positive or not. This is a study that leaves many questions unanswered.
Round 2
Reviewer 2 Report
I think this is a clinical study of a PET scan over a long period of 10 years, and I think there have been improvements in PET models over the 10 years. Does this mean that the detection power is the same even if the PET models are different?
Author Response
Thank you for your comments.
Point 1: I think this is a clinical study of a PET scan over a long period of 10 years, and I think there have been improvements in PET models over the 10 years. Does this mean that the detection power is the same even if the PET models are different?
Response 1: We think that improvements in PET models may have changed the detection power of lymph nodes. It may be a bias in our study. It has been added in the manuscript.
All the manuscript has been reviewed and corrected by a translator.